# The Current State and Future Prospects of *Auricularia auricula’s* Polysaccharide Processing Technology Portfolio

**DOI:** 10.3390/molecules28020582

**Published:** 2023-01-06

**Authors:** Te Yu, Qiong Wu, Bin Liang, Jiaming Wang, Di Wu, Xinzhu Shang

**Affiliations:** Department of Food Science and Engineering, College of Food Science and Engineering, Changchun University, No.6543, Satellite Road, Changchun 130022, China

**Keywords:** *Auricularia auricula*, polysaccharide, extraction technology, healthcare

## Abstract

*Auricularia auricula* polysaccharides (AAP) have been widely studied in the field of medicine and healthcare because of their unique structure and physiological activity. Many species of *Auricularia auricula* polysaccharides have been extracted, isolated, and purified by different methods, and their structures have been analyzed. *Auricularia auricula* polysaccharides have been proven to have beneficial effects on the human body, including slowing the aging process, controlling the intestinal system, and treating cardiovascular disorders. In this paper, the extraction, isolation, and purification of AAP from *Auricularia auricula,* as well as research in the field of medicine and healthcare, have pointed to the shortcomings and limitations of these methods. We also suggest future research directions for *Auricularia auricula* polysaccharides; standardized processing methods must be confirmed, and officially approved AAPs are needed for commercial applications. Finally, an optimistic outlook on the development of AAPs is given.

## 1. Introduction

*Auricularia auricula* is an edible mushroom and a traditional medicine in China, and it is also the fourth largest cultivated mushroom species in the world [1]. It is widely valued in Asia and around the world for its nutritional and healing properties. China, in particular, is a major producer of *Auricularia auricula*, which has been consumed for at least 4000 years and is used as a common traditional medicine in Asia [2,3].

*Auricularia auricula* is rich in polysaccharides, proteins, fats, vitamins, pigments, and trace elements [4]. *Auricularia auricula* polysaccharide (AAP) is the main active ingredient of *Auricularia auricula*, accounting for 60% of the total content [5,6]. In recent years, the research on AAPs in the field of healthcare has intensified, and a lot of chronic disorders have been found to respond quite favorably to its therapeutic effects [7].

Many studies have confirmed that polysaccharides from *Auricularia auricula* can regulate intestinal flora [8], enhance immune regulation [9], slow down the aging process [10,11], resist radiation [12], reduce blood lipids [13], and exterminate viruses [14]. These activities show great promise for the development of nutritional medicines and pharmaceuticals. The health benefits from AAP go far beyond the other essential nutrients in *Auricularia auricula,* and are a major direction for the future deep processing of *Auricularia auricula.*

Because the polysaccharide is a Fungi heteropolysaccharide, the uncertainty of its structure and composition has caused great difficulties for its in-depth study. The biological activity of the samples obtained by different extraction or drying methods was different [15]. Therefore, the study of extraction technology is the first hurdle in the in-depth study of the polysaccharides of *Auricularia auricula*. To improve the yield of polysaccharides, scientists have developed enzymatic extraction [16], ultrasonic extraction [17], liquid fermentation extraction [18], and other methods in recent years. As far as the polysaccharides obtained so far are concerned, four types of polysaccharides (A. corna (ACP), A. auricula (AAP), A. polytricha (APP), and M. murgy (MFP)) have mainly been studied [19]. Finding an extraction method with high reproducibility of the extracted sample structure and the molecular weight is a key research direction for the extraction of AAP. Moreover, the abovementioned traditional method of extracting *Auricularia auricula* polysaccharides leads to a large amount of cooking waste liquid and a large amount of nutrient loss. Recent studies have shown that the melanin in the residual species of the remaining *Auricularia auricula* has a great potential for resource exploitation, enabling the recultivation of other plant species on the waste and facilitating the preparation of beverages from whey protein isolates [20,21,22]. The reuse of *Auricularia auricula* residue will also be a future research direction.

China is currently the world’s largest producer of *Auricularia auricula*, accounting for more than 90% of the total global production. In 2018, the output of *Auricularia auricula* (dried products) reached 674,000 tons, with an output value of 37.46 billion yuan and a foreign exchange generation of 6.15 billion yuan. As a characteristic and advantageous agricultural product in China, mushrooms have played an important role in improving agricultural efficiency, increasing farmers’ incomes, and helping to alleviate poverty in the industry [20]. It is also important to note that the production scale of *Auricularia auricula* cultivation has increased year by year, and the market pressure on *Auricularia auricula* forest products has also been increasing with the popularization and application of bagging cultivation technology [21]. The cultivation scale of *Auricularia auricula* is expanding, the production is increasing year by year, and the profit margin of market sales is gradually shrinking. A gradual change from a seller’s market to a buyer’s market has been occurring. The deep processing of *Auricularia auricula* is one of the future development directions of *Auricularia auricula* products. Figure 1 shows the change in *Auricularia auricula* production in China from 2016 to 2022, which has been growing steadily since 2018, reaching 7,295,900 tons by 2020, an increase of 3.96% year-over-year [22]. By 2022, China’s *Auricularia auricula* production will grow to 7,723,400 tons, up 1.98% year-over-year [23]. With the continuous updating of cultivation technology and culture methods, further development of *Auricularia auricula* polysaccharide products and the deep processing of them is the way forward for the *Auricularia auricula* industry. As shown in Figure 2, among all *Auricularia auricula* deep processing products, *Auricularia auricula* beverage products have the largest market share at 28%, which is much higher than that for other products. The product with the second highest market share is *Auricularia auricula* nutritional powder, with an 8% market share followed by *Auricularia auricula* porridge and ready-to-eat *Auricularia auricula*, each with a 6% market share [22,24,25]. Therefore, in the Chinese market, *Auricularia auricula* is mainly used in products that help to lower blood lipids and enhance immunity.

The demand for polysaccharides as nutritional and functional components of *Auricularia auricula* is growing, and the market is expanding [26,27]. Many high-value (AAP) derivatives have been developed, and processing technologies have matured. Although they have many health benefits, they are not yet fully to market. *Auricularia auricula* polysaccharides have a unique structure with a wide range of biological and pharmacological activities. They are becoming increasingly popular worldwide due to their high content, ease of extraction, specific structure, minor side effects, and significant therapeutic potential. So far, many AAPs have been extracted from different A. nigra species distributed all over the world [11]. Numerous in-depth studies on the structure and biological and pharmacological activities of AAPs from different A. nigra species have been carried out. However, standardized processing methods have yet to be confirmed, and the commercial applications of AAP need to be officially approved. The aim of this review is to provide useful information and practical recommendations for the large-scale commercial production of AAP and to enable the rapid market entry of promising functional ingredients with economic value for the food industry and health benefits for consumers.

## 2. Pretreatment of *Auricularia auricula* and Extraction Technology

Pretreatment is the initial step before extracting polysaccharides from fungi. Its major objective is to eliminate undesired components while retaining polysaccharides. *Auricularia auricula* must be pretreated to obtain pure products. The protein and ash are frequently removed while treating *Auricularia auricula*. The related pretreatments of *Auricularia auricula* are provided in Table 1. The yields and physiological activities of polysaccharides obtained from different regions of the fungus using different extraction methods and different drying methods are very different.

To improve the commercial value of *Auricularia auricula* polysaccharides, the creation of a new grading standard is necessary. Investigating the effects of different molecular weights, different monosaccharide compositions, and different spatial structures on their physiological activities is of paramount importance.

*Auricularia auricula* polysaccharides account for 60% of its total content [28]. However, the yield of the current extraction process is not ideal, and improving the polysaccharide yield and enhancing its specific activity is the direction of further research on *Auricularia auricula* polysaccharides.

As far as the process of extracting polysaccharides is concerned, the raw material of *Auricularia auricula* is treated in two ways, as shown in Figure 3. The polysaccharides obtained by the two treatments were mainly composed of rhamnose, galactose, glucose, mannose, and xylose, but the molar ratios were different. The polysaccharides obtained from the fermentation broth of *Auricularia auricula* were mostly extracellular polysaccharides, which have better oxidation resistance, but the yield was lower than that of powdered raw materials [26,36].

We also found that the particle size of *Auricularia auricula* powder has a great influence on its extraction rate. The smaller the particle size, the higher the extraction rate. The intracellular polysaccharides of *Auricularia auricula* are largely restricted by the cell wall. The first step in extracting polysaccharides is to destroy the cell structure. This can be done by ultrasound, high temperature, or other methods, and the basic principle is to destroy the cell structure and increase the free constant of *Auricularia auricula* polysaccharide so as to obtain the target sample.

From the best conditions of these methods, we can infer several conclusions (except the enzyme method): (1) The smaller the particle size of comminution, the higher the extraction rate of polysaccharides; (2) The higher the extraction temperature and pressure, the higher the extraction rate of polysaccharides; (3) The longer the extraction time, the higher the extraction rate of polysaccharides. The effects of temperature, pressure, and time can be attributed to cell destruction theory. However, the particle size of *Auricularia auricula* powder is also related to its water absorption, referred to as “water absorption asymmetry of feather cells in polysaccharides”. This effect leads to the different stretching of polysaccharides and cell walls during hot water immersion, so the separation effect can be achieved. According to this principle, it can be inferred that the water absorption strength of *Auricularia auricula* may be related to the content of polysaccharides in the body. In order to further improve the extraction rate of *Auricularia auricula* polysaccharides, the cell wall should be destroyed as much as possible, and the particle size should be as large as the polysaccharide molecule.

In addition, according to our research, the cooking liquid obtained during the processing of *Auricularia auricula* is brown or dark black, and the protein content is high. The polysaccharides extracted from the body wall and cooking liquid of sea cucumber are brown. Therefore, it is suggested that the raw materials of the *Auricularia auricula* body wall should be pretreated to remove pigments and inorganic salts as much as possible. Otherwise, the obtained coarse AAP is dark brown with a high ash content. In addition, from the perspective of industrial applications, since organic solvents are not allowed to be used in the food and pharmaceutical industries, it is necessary to establish an alternative and effective protein removal method.

## 3. Separation and Purification

The significance of polysaccharide purification lies in the removal of small molecules, such as pigments and inorganic salts. Currently, separation methods include chromatography and dialysis. Purification methods can classify polysaccharides according to the criteria of molecular weight and moderate acidity. Because of their biological activity and non-toxic or low-toxicity properties, natural polysaccharides of food origin are gaining popularity in food, cosmetics, biomaterials, and other fields [37,38,39,40,41,42]. Purification also enables usable, safe, and reproducible polysaccharide research [43]. The choice of purification method is determined by the nature of the polysaccharide and is influenced by the manufacturing process. Impurities (starch, lipids, and pigments) that are not completely removed during raw material pretreatment may mix with polysaccharides during the subsequent polysaccharide extraction process. Furthermore, recent research has shown that different extraction conditions, such as pH and solvent temperature, can result in the permeation of different polysaccharides [44,45]. As shown in Figure 4, Wei et al. [40] divided the separation of polysaccharides into three stages: polysaccharides in raw material, polysaccharides in crude extract, and crude polysaccharides. The purification of crude polysaccharides includes the enrichment of polysaccharides and the separation of fractions with different structural or conformational characteristics.

Polysaccharides have been widely accepted as nutraceuticals, which improve the immune function of the body. However, many polysaccharides remain as health products and fail to be developed into drugs. The main reason for this is that the separation and purification of polysaccharides is difficult, and the current technology level does not meet the requirements for this. Generally speaking, polysaccharides are hydrophilic macromolecules. The separation and purification methods of polysaccharides are different from those of small molecules. In addition, different polysaccharides have different properties, so different separation and purification methods must be used. This work requires not only a theoretical knowledge of polysaccharides, but also accumulated working experience in the separation and purification of polysaccharides.

### 3.1. Concentration Grading Method

This method mainly takes advantage of the fact that the solubility of different polysaccharides in different concentrations of organic solvents is different. The solubility of polysaccharides with larger molecular weights in ethanol or acetone are less than it is of those with a smaller molecular weight [46]. Therefore, the molecular weight of the product can be controlled by adjusting the concentration of the organic solvent. This is usually done in the following manner:

While stirring, a high concentration of anhydrous ethanol is slowly added to the solution of the polysaccharide mixture to reach a final concentration of 25% ethanol (*v*/*v*). After the addition of ethanol, the solution is left for 2 h and then centrifuged to obtain the supernatant and precipitate (which may be referred to as the “first precipitate”). The precipitate is of a high MW polysaccharide grade. While stirring, ethanol is slowly added to the supernatant to reach a final concentration of 35% ethanol (*v*/*v*). The solution is left for 2 h and then centrifuged to obtain the supernatant and precipitate (which may be referred to as the “second precipitate”). The second precipitate is also a polysaccharide fraction, but its MW is lower than the first precipitate. The step-down process can be carried out further, depending on circumstances. The key to graded precipitation is to avoid co-precipitation as much as possible. The concentration of the polysaccharide mixture should not be too high, the ethanol should not be added too fast, and the pH of the solution should be near neutral. The lower the concentration of polysaccharide solution, the weaker the co-precipitation effect and the better the purification effect. However, if the polysaccharide concentration is too low, the recovery of the polysaccharide will be reduced and the consumption of ethanol will be greatly increased. Usually, the concentration of polysaccharide in the mixture is adjusted from 0.25% (*w*/*v*) to 3% (*w*/*v*) before using this method [47,48,49,50]. The concentration grading method is commonly used in the research and development of polysaccharide nutraceuticals because it is much easier than column chromatography.

### 3.2. Column Chromatography Method

Column chromatography is the most widely used method for the purification of polysaccharides. Several methods of column chromatography are described, as follows:

#### 3.2.1. Macroporous Resin

Macroporous adsorption resin is used to selectively adsorb organic substances from the solution by physical adsorption so as to achieve separation and purification. Its physical and chemical properties are stable; it is insoluble in acids, bases, and organic solvents, has good selectivity for organic substances, and it is unaffected by the presence of inorganic salts and strong ions, low molecular compounds, and swelling in water and organic solvents by the adsorption of solvents. The adsorption of macroporous resin relies on the van der Waals gravitational force between it and the adsorbed molecules (adsorbent), and works through its huge specific surface for physical adsorption, so that organic compounds can be separated by a certain solvent elution according to the adsorption force and its molecular weight size to achieve different purposes such as separation, purification, de-hybridization, and concentration. Macroporous resin can remove proteins, flavonoids, and pigments from polysaccharide solutions [51,52,53,54].

#### 3.2.2. Cellulose Column Chromatography

Cellulose is a common filling material in columns. First, after waiting for swelling, activation is performed using 0.5 mol/L NaOH with 0.5 mol/L HCL solution; the cellulose in the column is equilibrated with NaCL solution, and then the polysaccharide is loaded onto the cellulose column for purification. Afterwards, the cellulose columns are eluted separately using an eluent so that different polysaccharide levels can be continuously eluted [46]. Polysaccharides can be separated according to different molecular weights or acid-based groups. In the elution process, the various polysaccharide fractions undergo several dissolution and precipitation processes in the cellulose column, and can eventually be separated from each other. This method can be called the “graded dissolution method”, which is basically the opposite of the graded precipitation method. Due to the high number of theoretical plates in the cellulose column chromatography, the purity of the eluate is higher [55]. However, the disadvantage of this method is the low flow rate and the long period of time required. The flow rate seems to be too low, especially for highly viscous acidic polysaccharides.

#### 3.2.3. Gel Column Chromatography

Gel column chromatography is based on the size and shape of polysaccharide molecules, i.e., the molecular sieve principle, to separate polysaccharides. This chromatographic method is widely used for the separation and purification of polysaccharides. In general, the crude polysaccharides obtained are first purified using macroporous resin and cellulose chromatography, and are then further purified using gel column chromatography. Commonly used gels are various types of Sephadex, Sepharose, Bio gels, and later Sephacryl, Superdex, and Superose. The eluents are salt solutions and buffers of various concentrations.

## 4. Physiological Activity and Product Development of Polysaccharides of *Auricularia auricula*

### 4.1. Physiological Activity of Auricularia auricula

#### 4.1.1. Regulation of Intestinal Flora

The intestinal flora is essential for maintaining host health by regulating cellular activity and the immune system. According to previous studies, it is associated with leukemia infection [56], small bowel colitis [57], ischemic stroke [58], obesity [59], and a number of other harmful conditions.

According to the research, as shown in Figure 5, there are as many as 51 metabolites regulated by Auricularia auricula polysaccharide, which are mainly concentrated in the arginine biosynthesis pathway, followed by the arginine and proline, glycine, serine, and threonine, glycerophospholipids, and sphingolipid metabolic pathways [60]. These pathways can lower total and LDL cholesterol levels and alter the composition of the intestinal flora. The relative abundance levels of *Lactobacillus johnsonii*, *Weissella cibaria, Kosakonia covanii*, *Enterococcus faecalis*, *Bifidobacterium animalis*, and *Bacteroides uniformis* are significantly upregulated, while *Firmicutes* bacteria m10-2 are downregulated. The biological activity of AAP may be related to the regulation of the endogenous metabolism and intestinal flora composition. Zhang et al. [8]. found that A. auricular upregulated the high-abundance SCFA-producing genus *Bacteroides* and Paraprevotella in a dietary fiber-rich diet, while AAP could better enrich several lower-abundance SCFA-producing bacteria, such as *Flavonifractor* and *Clostridium IV*.

#### 4.1.2. Anti-High Cholesterol

*Auricularia auricula* polysaccharides have multiple regulatory effects on high cholesterol. AAP significantly reduces body lipid and triglyceride levels in *Cryptobacterium hidrad,* and has a significant protective effect against intracellular free radical generator-induced damage and increases the activity of antioxidant enzymes, including superoxide dismutase (SOD) and catalase (CAT) [13]. Using mice with hyperlipidemia as a model, AAP significantly reduced serum and liver TC, TG, and serum LDH-c levels in mice [61]. It can also protect the liver by enhancing antioxidant effects as a blood lipid-lowering agent [62].

#### 4.1.3. Hypoglycemic Effect

Hyperglycemia is also common globally. *Auricularia auricula* polysaccharide, as a botanical heteropolysaccharide, can reduce blood sugar, especially for streptozotocin-induced type 2 diabetes (T2DM) [63]. Its hypoglycemic activity is regulated by metabolic pathways. AAP can activate oxidative stress and NF-κB signaling and proinflammatory cytokine production [64], and regulates the akt/ampk signaling pathway [65]. AAP can control the blood sugar balance in the human body from multiple angles.

#### 4.1.4. Anti-Cancer

*Auricularia auricula* polysaccharides also play an important role in cancer treatment because of their safety and efficacy. For patients with gastrointestinal cancer (GIC), they significantly improved the treatment response rate and survival rate (0.5 years, 1 year, and 2 years), and improved immune function without increasing the incidence of adverse reactions. This treatment also has a good adjuvant effect on enhancing platinum (L-OHP and DDP) and adriamycin (ADM) [66]. In this context, therapies based on biopolymer prodrug systems represent promising alternatives to improve the pharmacokinetic and pharmacological properties of drugs and reduce their toxicity.

#### 4.1.5. Anti-Oxidation and Anti-Aging

Polysaccharides have a great effect on antioxidation because of their unique spatial structure, especially the *Auricularia auricula* polysaccharide.

The ABTS+ clearance of AAPs reached 37.95 ± 0.53% in Hidradenia [16]. Under acidic conditions, the clearance rate reached 97.94 ± 0.87% [67]. Different monosaccharide compositions and molecular weights also have different antioxidant effects. Therefore, in order to research the antioxidant mechanism and application in depth, polysaccharide raw materials should be divided and analyzed in greater detail.

#### 4.1.6. Anti-Viral

Polysaccharides have good antiviral activity. As an effective and low-toxic antiviral component, the polysaccharide has broad medical prospects and is worth further study.

The results showed that AAPS significantly inhibited the cell infectivity of NDV in the chicken embryo fibroblast (CEF) culture system [14].

### 4.2. Practical Application of Auricularia auricula Polysaccharide

#### 4.2.1. Biological Anticorrosive Film

Antibacterial and biocompatible films have attracted much attention due to their wide potential for application. Although a lot of work has been done in this area, the research in this field is still very active and is accompanied by the continuous development of new materials [68]. However, the research into polysaccharides in the field of biofilms is still in its early stages, and finding more suitable materials and process methods is the next step.

#### 4.2.2. Edible Products

*Auricularia auricula* has a history of thousands of years as a food and medicinal material, but its edible products are few and its audience is small, so it has no great commercial value. Currently, research on the consumption of *Auricularia auricula* is focused on its use as an ingredient in different food products. the complexation behaviour of AAP and whey protein isolates can be applied to the beverage industry, as exemplified by this [69].

## 5. Conclusions and Future Outlook

There are still some shortcomings and limitations in the pretreatment, extraction, separation, purification, and classification of *Auricularia auricula* polysaccharides. First, the current extraction, separation, purification, raw material pretreatment, and crude polysaccharide purification methods of *Auricularia auricula* polysaccharides remain in the laboratory stage, there is no standardized processing method, and commercial large-scale processing technology has yet to be established. The methods for raw material pretreatment and crude polysaccharide purification extracted from *Auricularia auricula* polysaccharides are under-researched. However, there is no doubt about the benefits of *Auricularia auricula* polysaccharides to human health. They can not only treat chronic diseases such as hyperglycemia and hyperlipidemia, but also have a good therapeutic effect in the treatment of serious diseases such as cancer.

## Figures and Tables

**Figure 1 molecules-28-00582-f001:**
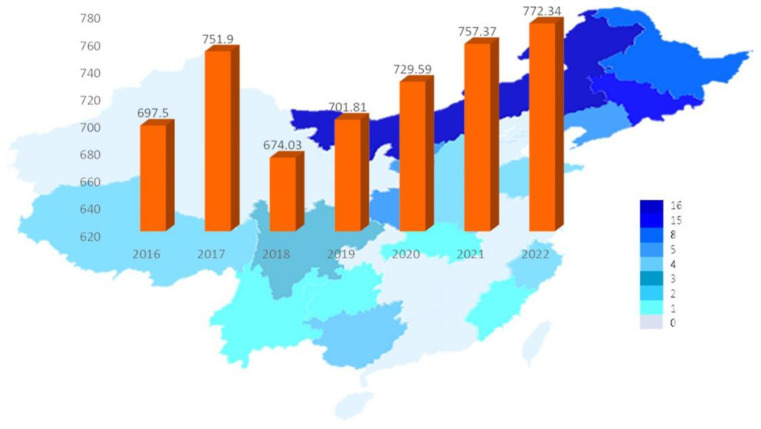
Production and forecast of *Auricularia auricula* in China from 2016–2022.

**Figure 2 molecules-28-00582-f002:**
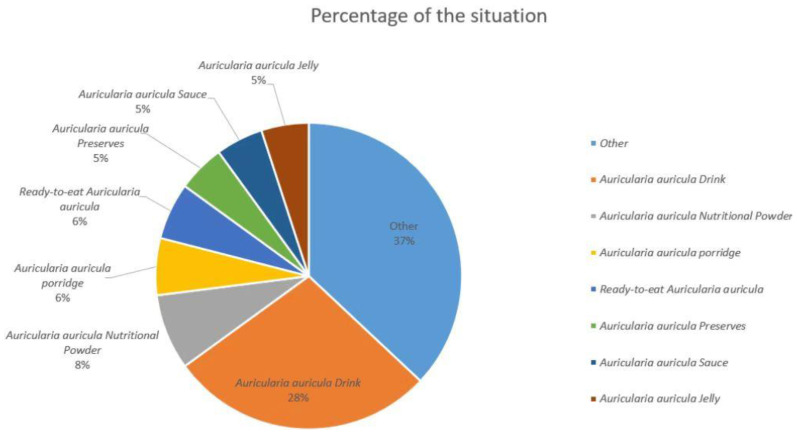
The proportion of commodity types in the *Auricularia auricula* market.

**Figure 3 molecules-28-00582-f003:**
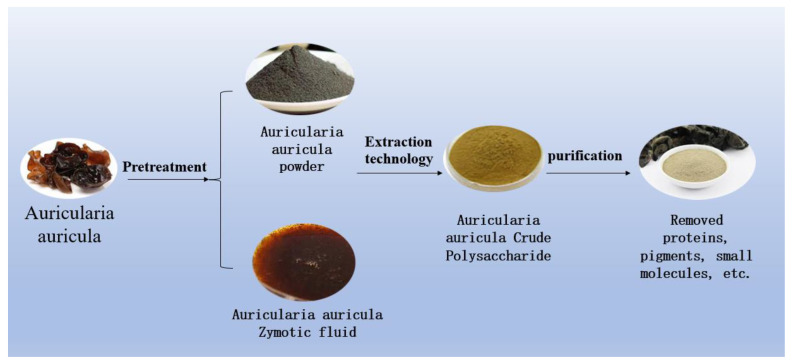
Two Methods of Processing Raw Materials of Auricularia auricula.

**Figure 4 molecules-28-00582-f004:**
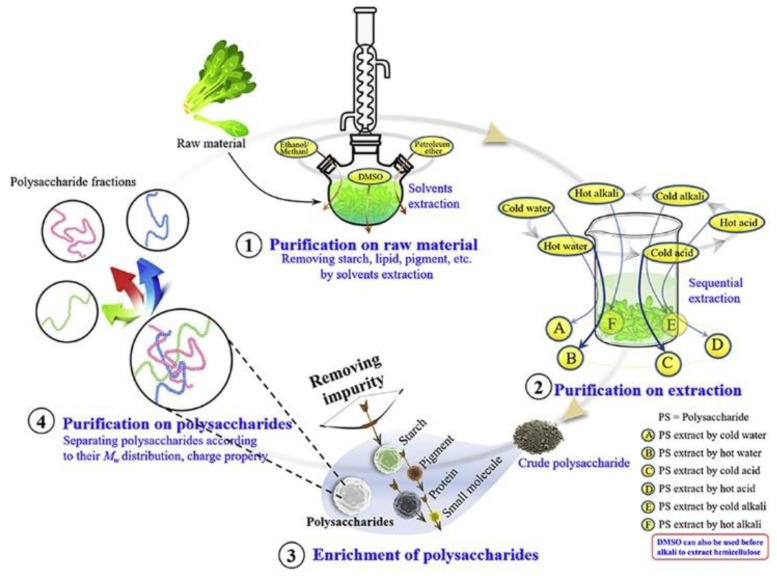
The purification process of polysaccharides.

**Figure 5 molecules-28-00582-f005:**
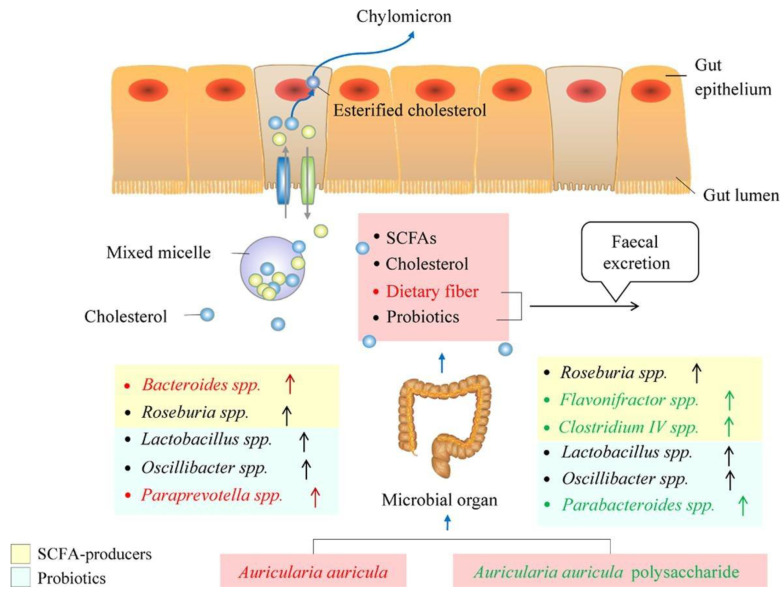
Mechanism of *Auricularia auricula* polysaccharides.

**Table 1 molecules-28-00582-t001:** Pretreatment, extraction, and separation of AAPs from different *Auricularia auricula*.

Origin	Extraction Method	Time (min)	Solid-Liquid Ratio	Temperature (°C)	Other Conditions	Yield (%)	References
Jinlin County, Jilin Province	Ultrasonic Assist	40	1:70	70 °C	Particle size of 150–200 mesh	29.29 ± 1.41%	[29]
Qingchuan County, Sichuan Province	MAE	25	1:25	95 °C	Microwave power of 860 W, pH 7.0	10.52%	[30]
Jiaohe County, Jilin Province	Pulsed Electric Field	1.5	1:30	Room temperature	HIPEF strength at 24 kV/cm, pulse number at 6, pH 8	14.79%	[31,32]
Jinan	PEG-based ultrasound-assisted	32.44	1:39.27	91.948 °C	PEG concentration of 0.30 g/mL	21.58%	[33]
Greater Khingan Mountains, Helongjiang	Neutral protease	/	1:75	50 °C	E/S at 8%	12.96%	[13]
Greater Khingan Mountains, Heilongjiang Province	Ultrasonic Assist Alkali method	90	1:48	70 °C	2.0% of NaOH concentration	15.53%	[34]
Greater Khingan Mountains, Heilongjiang Province	Mannanase, β-dextranase, and cellulase	60	1:80	83.17 °C	pH at 2.1	26.42% ± 0.87%	[35]

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
