# Peer review of "The Current State and Future Prospects of Auricularia auricula’s Polysaccharide Processing Technology Portfolio"

_molecules, 2023, doi:10.3390/molecules28020582_

Round 1
Reviewer 1 Report
The theme of the work is interesting, however it is necessary to improve the way of describing the theme, in order to clarify not only the biological origin of the polysaccharides under study, as well as their composition/structure, and extraction and purification processes.
- Species/genus names must be in italics
- “To improve the yield of polysaccharides, scientists have invented enzymatic extraction” change invented by developed
- Line 82 “aricularia auricula” change to Auricularia auricula
- At fig 1 what do the different shades of blue in the map mean?
- Black Cohosh polysaccharides are of plant or fungal origin?
- Line 97 “So far, 97 many AAPs have been extracted from different A. nigra species distributed all over the world35” the cited article is about the Antioxidant activity in vitro and in vivo of the polysaccharides from different varieties of Auricularia auricula
- Line 107 “Pretreatment is the initial step before extracting polysaccharides from plants and an- 107 imals. Its major objective is to eliminate undesired components while retaining polysac- 108 charides. Auricularia auricula must be pretreated to obtain pure products. The protein 109 and ash are frequently removed while treating Auricularia auricula” Auricularia auricular is a fungi. 8. Line 119 “improving the polysaccharide 119 yield and enhancing its specific activity is the direction of further research on Nigella sa- 120 tiva polysaccharides” what is the importance of Nigella for this study? 9. table 1 is unclear 10. The section on separation and purification is described in a very abbreviated way, which makes it difficult to understand the process. Besides, I don't understand why a plant appears in figure 4. The article” Nana Chen, Hao Zhang, Xin Zong, Siyu Li, Jiaojiao Wang, Yizhen Wang, Mingliang Jin,Polysaccharides from Auricularia auricula: Preparation, structural features and biological activities, Carbohydrate Polymers,Volume 247,2020,116750” explains this process in more detail
Author Response
Dear Editor:
I am the author of the article "Current status and prospect of polysaccharide processing technology portfolio of Auricularia auricula". Thank you very much for your suggestions on the revision of this article. Now we will report the revision of this article to you
1. Species/genus names must be in italics
Revisions have been made
2. To improve the yield of polysaccharides, scientists have invented enzymatic extraction” change invented by developed
Revisions have been made.To improve the yield of polysaccharides, scientists have developed enzymatic extraction
3. Line 82 “aricularia auricula” change to Auricularia auricula
Revisions have been made.The third is Auricularia auricula porridge and ready-to-eat Auricularia auricula, both with 6% market share
4. At fig 1 what do the different shades of blue in the map mean?
In Figure 1, the different shades of blue in the map imply the type of Auricularia auricula in each province of China
Number of local standards:
Inner Mongolia 16、Jilin 15、Heilongjiang 8、Liaoning 5、Shaanxi 5、Guangxi 4、Sichuan 3、Anhui 2、Henan 2、Shanxi 2、Xizang 2、Zhejiang 2、Shandong 2、Guizhou 1、Hubei 1、Yunnan 1、Fujian 1、Ningxia 0、Taiwan 0、Qinghai 0、Xinjiang 0、Hong Kong 0、Beijing 0、Hunan 0、Gansu 0、Hainan 0、Guangdong 0、Tianjin 0、Jiangxi 0、Jiangsu 0、Hebei 0、Shanghai 0、Macao 0、South China Sea Islands 0
I have completed the changes and explained in the figure and in the figure notes
Figure 1. Types、production and forecast of Auricularia auricula in China from 2016-2022.
5. Black Cohosh polysaccharides are of plant or fungal origin?
Black Cohosh polysaccharides are of Fungi of the genus Mucor, family Mucoraceae. Sorry for the misunderstanding, but the content has now been changed to Fungi of the genus Mucor, family Mucoraceae
6. Line 97 “So far, 97 many AAPs have been extracted from different A. nigra species distributed all over the world35” the cited article is about the Antioxidant activity in vitro and in vivo of the polysaccharides from different varieties of Auricularia auricula
The original text is “So far, many AAPs have been extracted from different A. nigra species distributed all over the world.” 97 is the line number. the cited article is about the Antioxidant activity in vitro and in vivo of the polysaccharides from different varieties of Auricularia auricula.
7. Line 107 “Pretreatment is the initial step before extracting polysaccharides from plants and an- 107 animals. Its major objective is to eliminate undesired components while retaining poly sac- 108 charides. Auricularia auricula must be pretreated to obtain pure products. The protein 109 and ash are frequently removed while treating Auricularia auricula” Auricularia auricular is a fungi.
Revisions have been made. Pretreatment is the initial step before extracting polysaccharides from Fungi. Its major objective is to eliminate undesired components while retaining polysaccharides. Auricularia auricula must be pretreated to obtain pure products.
8. Line 119 “improving the polysaccharide 119 yield and enhancing its specific activity is the direction of further research on Nigella sa- 120 tiva polysaccharides” what is the importance of Nigella for this study?
Revisions have been made. However, the yield of the current extraction process is not ideal, and improving the polysaccharide yield and enhancing its specific activity is the direction of further re-search on Auricularia auricula sativa polysaccharides
9. table 1 is unclear
Revisions have been made.
|
Origin |
Extraction Method |
Time (min) |
Solid-Liquid Ratio |
Temperature (◦C) |
Other Conditions |
Yield (%) |
References |
|
Jinlin County, Jilin Province |
Ultrasonic Assist |
40 |
1:70 |
70 ◦C |
Particle size of 150–200 mesh |
29.29 ± 1.41% |
[31] |
|
Qingchuan County, Sichuan Province |
MAE |
25 |
1:25 |
95 ◦C |
Microwave power of 860 W, pH 7.0 |
10.52% |
[32] |
|
Jiaohe County, Jilin Province |
Pulsed Electric Field |
1.5 |
1:30 |
Room temperature |
HIPEF strength at 24 kV/cm, pulse number at 6, pH 8 |
14.79% |
[33,34] |
|
Jinan |
PEG-based ultrasound-assisted |
32.44 |
1:39.27 |
91.948 |
PEG concentration of 0.30 g/mL |
21.58% |
[35] |
|
Greater Khingan Mountains, Helongjiang |
Neutral protease |
/ |
1:75 |
50 |
E/S at 8% |
12.96% |
[13] |
|
Greater Khingan Mountains, Heilongjiang Province |
Ultrasonic Assist Alkali method |
90 |
1:48 |
70 |
2.0% of NaOH concentration |
15.53% |
[36] |
|
Greater Khingan Mountains, Heilongjiang Province |
Mannanase, β-dextranase, and cellulase |
60 |
1:80 |
83.17 |
pH at 2.1 |
26.42% ± 0.87% |
[37] |
10. The section on separation and purification is described in a very abbreviated way, which makes it difficult to understand the process. Besides, I don't understand why a plant appears in figure 4. The article” Nana Chen, Hao Zhang, Xin Zong, Siyu Li, Jiaojiao Wang, Yizhen Wang, Mingliang Jin,Polysaccharides from Auricularia auricula: Preparation, structural features and biological activities, Carbohydrate Polymers,Volume 247,2020,116750” explains this process in more detail
The plant in the picture is just an example and not just for the purification of plant polysaccharides.
To introduce this section in more detail I have added the following:
Polysaccharides have been widely accepted as nutraceuticals to improve the immune function of the body. However, many polysaccharides only stay at the stage of health products and fail to be developed into drugs. The main reason is that the separation and purification of polysaccharides are difficult and the related technology level does not meet the requirements. Generally speaking, polysaccharides are hydrophilic macromolecules. The separation and purification methods of polysaccharides are different from those of small molecules. In addition, different polysaccharides have different properties, so different separation and purification methods must be used. This work requires not only theoretical knowledge of polysaccharides, but also accumulated working experience in the separation and purification of polysaccharides.
3.1.Concentration grading method
This method mainly takes advantage of the fact that the solubility of different polysaccharides in different concentrations of organic solvents is different. The solubility of polysaccharides with larger molecular weight in ethanol or acetone is less than that of those with smaller molecular weight50. Therefore, the molecular weight of the product can be controlled by adjusting the concentration of the organic solvent. This is usually done in the following manner:
While stirring, high concentration or anhydrous ethanol is slowly added to the solution of the polysaccharide mixture to reach a final concentration of 25% ethanol (v/v). After the addition of ethanol, the solution is left for 2 hours and then centrifuged to obtain the supernatant and precipitate (which may be referred to as the "first precipitate"). The precipitate is a high MW polysaccharide grade. While stirring, ethanol was slowly added to the supernatant to reach a final concentration of 35% ethanol (v/v). The solution is left for 2 hours and then centrifuged to obtain the supernatant and precipitate (which may be referred to as the "second precipitate"). The second precipitate is also a polysaccharide fraction, but its MW is lower than the first precipitate. The step-down process can be carried out further, depending on the circumstances. The key to graded precipitation is to avoid co-precipitation as much as possible. In fact, the concentration of the polysaccharide mixture should not be too high, the ethanol should not be added too fast, and the pH of the solution should be near neutral. The lower the concentration of polysaccharide solution, the weaker the co-precipitation effect and the better the purification effect. However, if the polysaccharide concentration is too low, the recovery of polysaccharide will be reduced and the consumption of ethanol will be greatly increased. Usually, the concentration of polysaccharide in the mixture is adjusted to 0.25% (w/v) to 3% (w/v ) before using this method51-54, Concentration grading method is commonly used in the research and development of polysaccharide nutraceuticals because it is much easier than column chromatography.
3.2Column chromatography method
Column chromatography is the most widely used method for the purification of polysaccharides. Several methods of column chromatography are described as follows:
3.2.1 Macroporous resin
Macroporous adsorption resin is used to selectively adsorb organic substances from solution by physical adsorption, so as to achieve the purpose of separation and purification. Its physical and chemical properties are stable, insoluble in acids, bases and organic solvents, good selectivity for organic substances, unaffected by the presence of inorganic salts and strong ions, low molecular compounds, and swelling in water and organic solvents by adsorption of solvents. The adsorption of macroporous resin relies on the van der Waals gravitational force between it and the adsorbed molecules (adsorbent), and works through its huge specific surface for physical adsorption, so that organic compounds can be separated by a certain solvent elution according to the adsorption force and its molecular weight size to achieve different purposes such as separation, purification, de-hybridization and concentration.Macroporous resin can remove proteins , flavonoids and pigments from polysaccharide solutions55-58.
3.2.2 Cellulose column chromatography
Cellulose is a common filling material in columns. First, after waiting for swelling, activation is performed using 0.5 mol/L NaOH with 0.5 mol/L HCL solution, then the cellulose in the column is equilibrated with NaCL solution, and then the polysaccharide is loaded onto the cellulose column for purification. Afterwards, the cellulose columns were eluted separately using eluent so that different polysaccharide levels could be continuously eluted59. It can separate polysaccharides according to different molecular weight or acid-base groups. In the elution process, the various polysaccharide fractions undergo several dissolution and precipitation processes in the cellulose column and can eventually be separated from each other. This method can be called "graded dissolution method", which is basically the opposite of graded precipitation method. Due to the high number of theoretical plates in the cellulose column chromatography, the purity of the eluate is higher60. However, the disadvantage of this method is the low flow rate and the long time required. Especially for highly viscous acidic polysaccharides, the flow rate seems to be too low.
3.2.3 Gel column chromatography
Gel column chromatography is based on the size and shape of polysaccharide molecules, i.e., the molecular sieve principle, to separate polysaccharides. This chromatographic method is widely used for the separation and purification of polysaccharides. In general, the crude polysaccharides obtained are first purified using, macroporous resin and cellulose chromatography, and then further purified using gel column chromatography. Commonly used gels are various types of Sephadex, Sepharose, Bio gels, and later Sephacryl, Superdex and Superose. The eluents are salt solutions and buffers of various concentrations.
11. There is a problem with the language and layout of the article
I have conducted English editing
We deeply appreciate your consideration of our manuscript, and we look forward to receiving comments from the reviewers

Reviewer 2 Report
The current manuscript reports the current state and future prospects of Auricularia auricula's polysaccharide processing technology portfolio.
The submitted manuscript is of low quality and poorly structured, some sections provide insufficient discussion.
You should proofread the text well, replace speech turns on your own behalf (lines 130, 137, 143)
In the Introduction, “Auricularia auricularia auricula” appears several times, is this not a mistake?
Table 1 is not displayed correctly
Sections 2 and 3 could be merged and separated into subsections similar to section 4.
Subsections 4.2-4.6 are very short, it may not make sense to separate them, but simply highlight the effect on the body in the text, or expand these subsections. The same remarks for section 5.
Author Response
1. Auricularia auricularia auricula” appears several times
Revisions have been made.
2. Sections 2 and 3 could be merged and separated into subsections similar to section 4.Subsections 4.2-4.6 are very short, it may not make sense to separate them, but simply highlight the effect on the body in the text, or expand these subsections. The same remarks for section 5
According to your suggestion I have expanded the third part of the purification content (NO.10), combining the content of the fourth and fifth parts into a new paragraph
- Physiological activity and product development of polysaccharides of Auricularia auricula
4.1. Physiological activity of Auricularia auricula
4.1.1Regulation of intestinal flora
Intestinal flora is essential for maintaining host health by regulating cellular activity and the immune system in the body. According to previous studies, it is associated with leukemia infection61, small bowel colitis62, ischemic stroke63, obesity64, and a number of other harmful conditions.
According to the research, there are as many as 51 metabolites regulated by Auricularia auricula polysaccharide, which are mainly concentrated in the arginine biosynthesis pathway, followed by arginine and proline; Glycine, serine and threonine and glycerophospholipids, as well as sphingolipid metabolic pathways65. It can reduce the levels of total cholesterol and low-density lipoprotein cholesterol, and change the composition of intestinal flora. The relative abundance levels of Lactobacillus johnsonii, Weissella cibaria, Kosakonia covanii, Enterococcus faecalis, Bifidobacterium animalis, and Bacteroides uniformis were significantly upregulated, while the Firmicutes bacteria m10-2 were downregulated66. The biological activity of AAP may be related to the regulation of endogenous metabolism and intestinal flora composition. Zhang et al67. found that A. auricular upregulated the high-abundance SCFA-producing genus Bacteroides and Paraprevotella related to a dietary fiber-rich diet, while AAP could better enrich several lower-abundance SCFA-producing bacteria such as Flavonifractor and Clostridium IV.
Figure 5. Mechanism of Auricularia auricula polysaccharides.
4.1.2 Anti-high cholesterol
Auricularia auricula polysaccharide has multiple regulatory effects on high cholesterol. AAP significantly reduced body lipid and triglyceride levels in Cryptobacterium hidrad had a significant protective effect against intracellular free radical generator-induced damage and increased the activity of antioxidant enzymes, including superoxide dismutase (SOD) and catalase (CAT)68. Using mice with hyperlipidemia as a model, AAP significantly reduced serum and liver TC, TG, and serum LDH-c levels in mice69. It can also protect the liver by enhancing antioxidant effects, as a blood lipid-lowering agent70
4.1.3. Hypoglycemic effect
Hyperglycemia is also common in the current world. Auricularia auricula polysaccharide, as a botanical heteropolysaccharide, can reduce blood sugar, especially for streptozotocin-induced type 2 diabetes (T2DM)71. Its hypoglycemic way is regulated by metabolic pathways. AAP can activate oxidative stress and NF- κ B signaling and proinflammatory cytokine production72 and regulates akt/ampk signaling pathway73. It can be said that AAP can control the blood sugar balance in the human body from multiple angles
4.1.4. Anti-cancer
Auricularia auricula polysaccharide also plays an important role in cancer treatment because of its safety and efficacy. For patients with gastrointestinal cancer (GIC), it significantly improved the treatment response rate and survival rate (0.5 years, 1 year, and 2 years), and improved immune function without increasing the incidence of adverse reactions. It also has a good adjuvant effect on enhancing platinum (L-OHP and DDP) and adriamycin (ADM)74. In this context, therapies based on biopolymer prodrug systems represent promising alternatives to improve the pharmacokinetic and pharmacological properties of drugs and reduce their toxicity.
4.1.5. Anti-oxidation and anti-aging
Polysaccharide has a great effect on antioxidation because of their unique spatial structure, especially the Auricularia auricula polysaccharide.
It was shown that the ABTS+ clearance of AAPs could reach 37.95 ± 0.53% in Hidradenia.17, Under acidic conditions, the clearance rate can even reach 97.94 ± 0.87%75. Different monosaccharide compositions and molecular weights also have different antioxidant effects 68. Therefore, in order to research its antioxidant mechanism and application in depth, polysaccharide raw materials should be divided and analyzed in more detail
4.1.6. Anti-viral
Polysaccharide has good antiviral activity. As an effective and low-toxic antiviral component, the polysaccharide has a broad medical prospect and is worth further study.
The results showed that AAPS significantly inhibited the cell infectivity of NDV in the chicken embryo fibroblast (CEF) culture system76
4.2. Practical application of Auricularia auricula polysaccharide
4.2.1 Biological anticorrosive film
Antibacterial and biocompatible films have attracted much attention due to their wide applications. Although a lot of work has been done in this area, the research in this field is still very active and is accompanied by the continuous development of new materials.77 However, the research of polysaccharides in the field of biofilm is still a big problem, and finding better suitable materials and process methods is the next step
4.2.2 Edible products
Auricularia auricula has a history of thousands of years as a food and medicinal material, but its edible products are single and its audience is small, so it has no great commercial valueAt present, the edible research of Auricularia auricula is mainly to add it to different foods as a raw material. For example, the complexation behavior of AAP and whey protein isolate can be applied in the beverage industry78
3. There is a problem with the language and layout of the article
I have conducted English editing
4. table 1 is unclear
Revisions have been made.
|
Origin |
Extraction Method |
Time (min) |
Solid-Liquid Ratio |
Temperature (◦C) |
Other Conditions |
Yield (%) |
References |
|
Jinlin County, Jilin Province |
Ultrasonic Assist |
40 |
1:70 |
70 ◦C |
Particle size of 150–200 mesh |
29.29 ± 1.41% |
[31] |
|
Qingchuan County, Sichuan Province |
MAE |
25 |
1:25 |
95 ◦C |
Microwave power of 860 W, pH 7.0 |
10.52% |
[32] |
|
Jiaohe County, Jilin Province |
Pulsed Electric Field |
1.5 |
1:30 |
Room temperature |
HIPEF strength at 24 kV/cm, pulse number at 6, pH 8 |
14.79% |
[33,34] |
|
Jinan |
PEG-based ultrasound-assisted |
32.44 |
1:39.27 |
91.948 |
PEG concentration of 0.30 g/mL |
21.58% |
[35] |
|
Greater Khingan Mountains, Helongjiang |
Neutral protease |
/ |
1:75 |
50 |
E/S at 8% |
12.96% |
[13] |
|
Greater Khingan Mountains, Heilongjiang Province |
Ultrasonic Assist Alkali method |
90 |
1:48 |
70 |
2.0% of NaOH concentration |
15.53% |
[36] |
|
Greater Khingan Mountains, Heilongjiang Province |
Mannanase, β-dextranase, and cellulase |
60 |
1:80 |
83.17 |
pH at 2.1 |
26.42% ± 0.87% |
[37] |
We deeply appreciate your consideration of our manuscript, and we look forward to receiving comments from the reviewers.

Round 2
Reviewer 1 Report
Please just change in figure Types for varieties
Reviewer 2 Report
Dear Authors, thank you for the corrections.